# Predominant Mycotoxins, Pathogenesis, Control Measures, and Detection Methods in Fermented Pastes

**DOI:** 10.3390/toxins12020078

**Published:** 2020-01-23

**Authors:** Guozhong Zhao, Yi-Fei Wang, Junling Chen, Yunping Yao

**Affiliations:** 1State Key Laboratory of Food Nutrition and Safety, Key Laboratory of Food Nutrition and Safety, Ministry of Education, College of Food Science and Engineering, Tianjin University of Science & Technology, 300457 Tianjin, China; zhaoguozhong@tust.edu.cn (G.Z.); yifeiwang@mail.tust.edu.cn (Y.-F.W.); 2College of Food and Bioengineering, Henan University of Science and Technology, 471023 Luoyang, China; chenjl2020@haust.edu.cn

**Keywords:** mycotoxins, fermented paste, pathogenicity, control measures, detection

## Abstract

Fermented pastes are some of the most popular traditional products in China. Many studies reported a strong possibility that fermented pastes promote exposure to mycotoxins, including aflatoxins, ochratoxins, and cereulide, which were proven to be carcinogenic and neurotoxic to humans. The primary mechanism of pathogenicity is by inhibiting protein synthesis and inducing oxidative stress using cytochrome P450 (CYP) enzymes. The level of mycotoxin production is dependent on the pre-harvest or post-harvest stage. It is possible to implement methods to control mycotoxins by using appropriate antagonistic microorganisms, such as *Aspergillus niger*, *Lactobacillus plantarum*, and *Saccharomyces cerevisiae* isolated from ordinary foods. Also, drying products as soon as possible to avoid condensation or moisture absorption in order to reduce the water activity to lower than 0.82 during storage is also effective. Furthermore, organic acid treatment during the soaking process reduces toxins by more than 90%. Some novel detection technologies based on magnetic adsorption, aptamer probes, and molecular-based methods were applied to rapidly and accurately detect mycotoxins in fermented pastes.

## 1. Introduction

Fermented pastes are very popular in China and include soybean paste, thick broad-bean paste, flour paste, and chili paste. The unique flavor, taste, and nutritional components of the paste products mainly depend on the long-term fermentation by aroma-producing microorganisms [1]. The antioxidant activities of fermented pastes may be improved through prolonged fermentation [2]. However, several undesired mycotoxins may exist and can lead to human disease due to the high risk of paste contamination during the long-term fermentation process [3]. Mycotoxins produced by fungi play the most significant role in food contamination [4]. For instance, high levels of mycotoxins were detected in wheat, coffee, grape products, and even in animal feed [5]. Some fermented products were reported to contain an abundance of *Aspergillus*, *Penicillium*, and *Fusarium* spp., which are correlated with the presence of aflatoxins or ochratoxin A [6].

It is well known that mycotoxin contamination in food has a huge risk to human health by suppressing the immune system, leading to conditions such as nephrotoxicity and teratogenesis. Most mycotoxins are regarded as potent carcinogens by the International Agency for Research on Cancer (IARC) [7]. Previous researchers investigated the pathogenesis of mycotoxins, which are associated with alterations in the epigenetic state involving some primary enzymes [8]. In general, most mycotoxins possess properties such as stability, low molecular weight, and heat resistance, thereby making food contamination much more difficult to prevent [9].

The main objective of this review is to summarize the potential mycotoxins, including aflatoxins, ochratoxins, and cereulide, that can exist in fermented pastes. We also explore the significant risk to human health and the pathogenic mechanisms, including carcinogenicity, teratogenicity, nephropathy, and neurotoxicity. Furthermore, the most novel technologies to control and detect different toxins in fermented products are also discussed.

## 2. Mycotoxins in Fermented Pastes

### 2.1. Aflatoxins

Aflatoxins are ubiquitously found in cereals, milk, tree nuts, and oilseeds and are considered to make up a group of extremely toxic metabolites produced by *Aspergillus flavus*, *Aspergillus parasiticus*, and *Aspergillus nomius* during almost all stages of paste production [10]. Previous research demonstrated that there more than 20 aflatoxins were present in pods and soybean seeds [11]. Aflatoxins slightly dissolve in water and are heat-stable [12]. The major aflatoxins include the B-types, i.e., aflatoxin B_1_ (AFB_1_) and aflatoxin B_2_ (AFB_2_), which are produced from the fusion of bifurano coumarins to cyclopentanone, and the G-types, i.e., aflatoxin G_1_ (AFG_1_) and aflatoxin G_2_ (AFG_2_), which are produced from the fusion of bifurano coumarins to lactone [13]. Efficient aflatoxin-producing species of the *Aspergillus* section Flavi were recently described (Table 1) [14]. Extensive investigations were performed, showing that aflatoxins are polyketide-derived secondary mebabolites produced via the conversion path of acetate polyketide to anthraquinones (norsolorinic acid, averantin, averufanin, averufin, and versicolorin A) to xanthones to aflatoxins (Figure 1) [15].

The production of aflatoxins during paste fermentation is affected by fermentation conditions, including the protein ratio of raw materials, starter cultures, and so on [24]. *A. flavus* and its close relatives are commensal with the crop and infect it during growth to produce aflatoxins before harvest. Some saprophytic species contaminate crops only after harvest [25]. Previous researchers showed that the growth conditions of these saprophytic species were above 37 °C and close to 0.80 water activity [26].

Crops contaminated by different *Aspergillus* species demonstrate various levels of risk regarding the presence of aflatoxins in final fermented pastes. For example, small grains like soybean and wheat do not have affinity with *Aspergillus* species, and aflatoxins can only be produced during untimely drying or storage at high- temperatures after harvest [27]. Crops such as peanuts and maize have higher risk levels than others, because *A. flavus* and *A. parasiticus* are commensal with these crops [28]. Contaminated soil further promotes mold on the plants if unharvested grains are not cleaned up in time [29]. Aflatoxins are not be degraded during the fermented process [30].

### 2.2. Ochratoxins

Ochratoxins are secondary metabolites produced by *Penicillium* or *Aspergillus* fungi under diverse conditions, with 18 related isomers, of which ochratoxin A (OTA) is the most toxic and considered to be teratogenic, neurotoxic, nephrotoxic, and carcinogenic in humans [31,32]. OTA contamination occurs in various foods, including soybeans, dried fruits, and coffee, and was also detected in fermented wines and grape juices [33]. Furthermore, the biosynthetic pathway of OTA in fungi was confirmed to be strongly related to polyketide synthase (otapksPN) and non-ribosomal peptide synthetase (npsPN), which are regarded as the primary enzymes [34]. Previous research demonstrated that there were three steps of OTA biosynthesis, namely, polyketide synthesis, acyl activation, and de-esterification by esterase [35].

The occurrence of OTA in fermented pastes is influenced by many factors, including the type of crop, the farming practice, the machining process, and environmental conditions. Previous studies showed that the prevalence of OTA in maize was very limited compared with other crops [36]. OTA was detected in 34.2% of wheat samples in six provinces of China [37]. Unlike aflatoxins, ochratoxin-producing pathogens are mostly regarded as post-harvest spoilage. For instance, *Penicillium* species were confirmed to be strongly correlated with stored cereals, such as soybeans, beans, and green coffee [38], therefore, the storage and process conditions were found to greatly influence the final products. Moisture is also a critical factor in the contamination of OTA during storage; the safe-storage water activity level is below 0.7 [39] and the risk of OTA occurrence is increased if the drying process is not rapid and moisture is absorbed on the surface of products in cold climates. Damage to the raw materials during harvest, processing, and storage is also a significant factor [40].

### 2.3. Cereulide

*Bacillus cereus*, a spore-forming, poisonous, Gram-positive bacteria, is especially relevant for immunocompromised patients, drug users, newborns, and post-surgical patients. Strains isolated from parenteral infections are able to synthesize necrotizing exotoxin-like phospholipase and hemolysin [41]. *B. cereus* can be isolated from human dentin plaques, but accounts for a small proportion of oral infections. Collegenase-producing Soc 67 and Ply 19 were isolated from plaque in a patient with gingivitis and a teenager with periodontitis [42].

*Bacillus cereus* can be isolated from a variety of fermented foods and environments, such as doenjang and soybean paste. *B. cereus* phages were isolated from 47 traditional Korean fermented foods, as well as 63.6% of doenjang samples, 100% of cheonggukjang samples, and 40% of gochujang samples [43]. Cereulide is considered to be an important foodborne toxin during paste fermentation due to its ability to cause diarrheal (heat-labile) and emetic (heat-stable) food poisoning after ingestion of more than 10^4^ Colony-Forming Units (CFU) per gram of food [44,45]. Controlling cereulide in in industrial practices is difficult due to the wide range of temperature (5–55 °C) in which *B. cereus* can grow [46].

## 3. Pathogenicity Mechanism of Mycotoxins

### 3.1. Cancer Diseases

A large amount of research regarding the toxicological mechanisms of mycotoxins revealed that many foodborne mycotoxins inhibit protein synthesis and induce oxidative stress, thereby causing various cancers [47]. For instance, fermented pastes contaminated by AFB_1_ cause latent damage, leading to human hepatocellular carcinoma (HCC), a common liver cancer. Previous research confirmed that the metabolic activation of AFB_1_ had a strong correlation with cytochrome P450 (CYP) enzymes, including CYP_1_A_2_, and CYP_3_A_4_, which are mainly expressed in the human liver [48]. HCC is metabolized to an AFB_1_-epoxide, and then adducts into AFB_1_–DNA [49], induces the transversion of G to T within codon 249 of the tumor suppressor gene p53, which is associated with liver cancer [9].

Furthermore, OTA was reported to be a group 2B human carcinogen by the International Agency for Research on Cancer (IARC) [32]. Due to the nephrotoxicity of OTA, ingesting products abundant in OTA increases the risk of suffering from Balkan Endemic Nephropathy (BEN) and Tunisian Nephropathy (TCIN) [50]. Recent research revealed that oxidative stress induced by OTA in HK-2 cells regulated the translocation of the transcription factors aryl hydrocarbon receptor (AhR) and pregnane X receptor (PXR) [51]. Activated AhR leads to immunosuppression and cancer by regulating CYP_1_A_1_ and CYP_1_A_2_ enzymes, and activated PXR has a relationship with CYP_3_A_4_, playing a role in phase I metabolism. Furthermore, activated Nrf2 inhibits cell death and decreases the expression of AhR and PXR in the presence of OTA, resulting in reduced renal damage [52]. Therefore, the regulation of transcription factors Nrf2, PXR, and AhR could help to treat and prevent renal injury by OTA (Figure 2) [53].

### 3.2. Neurodegenerative Diseases

Alongside carcinogenicity, OTA was demonstrated to impact migration, neuronal proliferation, and DNA content, thereby leading to neurodegenerative diseases and brain dysfunction of human such as parkinsonism and Alzheimer’s disease [54]. The neurotoxicity of the hippocampus, ventral mesencephalon, and striatum is more pronounced than in the cerebellum [55]. The mechanism of neurotoxicity involves OTA causing acute depletion of striatal dopamine and decreasing the immunoreactivity of striatum tyrosine hydroxylase, alongside a transient inhibition of DNA oxidative repair activity (oxyguanosine glycosylase, OGG_1_) [56]. Furthermore, OTA treatment inhibits the activation of SH-SY5Y, caspase-9, and caspase-3, which is accompanied by a decrease in mitochondrial membrane potential [57].

### 3.3. Gastrointestinal Diseases

Diarrhea, a type of gastrointestinal syndrome induced by ingesting *B. cereus*-contaminated fermented pastes, is attributed to enterotoxins and virulence factors including cytotoxin K (CytK), hemolysin BL (HBL), nonhemolytic enterotoxin (NHE), and enterotoxin FM (EntFM) [58]. CytK, Nhe, and HBL cause osmotic lysis of intestinal epithelial cell membranes due to hemolytic and cytotoxic activity [59]. Both HBLand Nhe are pore-forming toxins that cause intestinal fluid secretion [60]. HBL is a three component enterotoxin encoded by the hblA, hblD, and hblC genes, respectively [61]. CytK, which is encoded by the cytK gene, is more prevalent than the Nhe and HBL complexes (Figure 3) [62].

### 3.4. Emetic Illness

The emetic type of foodborne illness is induced by the small, cyclic, heat-stable toxin cereulide, which causes vomiting and nausea. [63]. The *ces* gene encodes cereulide synthetase, which is associated with the emetic toxin-producing strain of *B. cereus* [64]. Cereulide destroys the electrochemical gradient of membranes and inhibits fatty acid metabolism in mammalian cells by uncoupling mitochondrial ATP synthesis and disrupting the mitochondrial membrane potential [65]. Furthermore, the biological activity of the cereulide toxin impairs mitochondrial functionality (Figure 3) [66].

The pathogenicity mechanism of mycotoxins is briefly listed in Table 2. 

## 4. Methods to Control and Manage Mycotoxins

### 4.1. Biocontrol Method

Biocontrol is one of the most effective techniques of controlling various mycotoxins; it uses safe microorganisms in fermented pastes to inhibit pathogen growth [67]. The advantage of the biocontrol method is that the growing conditions of related antagonistic microorganisms are simple enough, and they have strong abilities to multiply in foods for long periods of time [68]. In addition, unlike antagonists, they do not carry the risk of producing allergenic spores, which is harmful to the environment and human health [69].

Several works showed that many lactic acid bacteria (LAB) strains were able to restrict the growth of *A. flavus*. For instance, both *L. plantarum* and related low-molecular-weight substances showed significant antifungal activity [70]. Recent research revealed that various novel metabolites of LAB, including 13-hydroxy-9-octadecenoic acid, 10-hydroxy-12-octadecenoic acid [71], phenolic antioxidants [72], and spermine-like and short cyclic polylactates [73], showed lower minimum inhibitory concentrations of mycotoxins compared to the ordinary organic acids. *Lactobacillus fermentum* YML014 was also shown to reduce *A. flavus* by up to 50% after isolation from Nigerian fermented food (Cassava) [74].

The mechanism of LAB against fungal mycotoxins involves metabolic conversion by binding the related mycotoxin to the cell wall through adsorption and enzymatic degradation [75]. The coexistence of biocontrol agents and mycotoxins in the environment result in competition for nutrients and space, thereby directly impacting the growth of various mycotoxins. In addition, some biocontrol strains modify the external environment to inhibit mycotoxin production. For instance, LAB-induced decrease in pH led to a reduction in aflatoxin production by 91% by decreasing the stability of related mycotoxins [76]. Furthermore, oxidative stress induced by LAB was also responsible for the production of mycotoxins. It was reported that aflatoxins were produced under oxidative stress conditions, so antioxidants could play a restoration role [77]. The other widely accepted mechanism is that mycotoxins bind to LAB cell wall compounds through absorption. Polysaccharides and peptidoglycans are regarded as the most important components to bind OTA to *L. plantarum* and *Lactobacillus sanfranciscensis*, which could be enhanced through genetic modification [78]. Unlike OTA, aflatoxins were demonstrated to bind peptidoglycans and teichoic acids of the *Lactobacillus rhamnosus* cell wall; this binding ability was decreased by heat-treating [79].

Moreover, the mechanism of mycotoxin regulation by *Aspergillus niger* FS10 was also studied. The results confirmed that the culture filtrate of *A. niger* FS10 had a strong ability to disrupt spore morphology, deform the cell wall, and restrict spore germination [80]. Also, the degradation rate of AFB_1_ in the culture filtrate was 85.5%, which showed the feasibility of applying *A. niger* FS10 culture filtrate to control AFB_1_ contamination in fermented pastes.

*Saccharomyces cerevisiae* is one of the most important yeasts to produce various volatile aroma compounds, including acetate ester, which plays a significant role in the flavor formation of fermented pastes [81]. *S. cerevisiae* was revealed to be a potential biocontrol agent. For example, both *S. cerevisiae* EBF101 and *S. cerevisiae* 117 were proven to inhibit the growth of *A. flavus* Z103 by up to 85% and 83%, respectively [82]. Moreover, 1,2-benzenedicarboxylic acid dioctyl ester, which is one of the primary metabolites of *S. cerevisiae*, possesses antifungal and antibacterial properties (Figure 4).

### 4.2. Physicochemical Control Methods

Different grains make up fermented pastes. Various *Aspergillus* fungi strains generally exist after long-term grain storage. Hull-less crops have higher levels of AFB_1_ accumulation than hulled crops during storage [83]. Environmental conditions, such as temperature and water activity (A_W_), were confirmed to affect the production of mycotoxins by inhibiting the expression of related genes [84]. Regardless of temperature, AFB_1_ could not be generated at an A_W_ of 0.82 [85]. Furthermore, the optimal temperatures of mycotoxin production ranged from 25 to 30 °C [86]. In general, temperature and A_W_ are considered to be the most important factors in the management of crop storage. By controlling the A_W_ to be stable and lower than 0.82 and the temperature to be lower than 37 °C, mycotoxins can be effectively inhibited at the source.

Most fermented pastes in China are made up of soybeans, which need to be soaked for 6–18 h in water. Adding organic acids (1 N citric acid and tartaric acid) during the soaking process reduced AFB_1_ by 94.1% and 95.1%, respectively [87]. It is well known that AFB_1_ is converted into β-keto acid and then to AFD_1_, which can be hydrolyzed [88]. Organic acids cannot impact the characteristics of various materials. Acidic or alkaline pH levels were shown to be effective in reducing the total amount of aflatoxins. Also the process of autoclaving significantly decreased the level of AFB_1_ at different pH levels compared to non-autoclaved samples [87] (Figure 4).

## 5. Detection Methods of Mycotoxins

At present, the main technologies regarding the determination of mycotoxins are based on chromatography, such as HPLC [89] and LC-MS/MS [90]. The methods of sample extraction include magnetic solid-phase extraction (MSPE) [91], solid-phase extraction (SPE) [92], and liquid–liquid extraction (LLE) [93]. Immunoassays are widely accepted to be able to detect mycotoxins sensitively in a variety of foods and feeds [94]. However, the techniques mentioned above include complicated and separated processes. Many novel methods exist which are simpler and more accurate.

### 5.1. Novel Magnetic Adsorbent Techniques

MSPE is a widely used method involving magnetic adsorption, whereby a product can be isolated from a sample solution using a magnet after the adsorption of analytes [95]. Recent studies focused on a new carbon-based nanomaterial named graphene as a magnetic adsorbent, which possesses unique properties such as strong adsorption capacity and excellent chemical stability. Furthermore, 3D graphene (3D-G), which has a low-density porous structure and versatile functionalization, overcomes the shortcomings of 2D graphene (2D-G) [96].

A new type of magnetic SPE, named graphene-coated magnetic-sheet SPE, was developed, in which magnetic 3D-G@Fe_3_O_4_ nanoparticles fixed on a perforated magnetic sheet were placed in a syringe filter holder and used as the adsorbent phase for the first time [97]. This technique was used to extract AFB_1_, B_2_, G_1_, and G_2_ rapidly, i.e., within 30 min, from fermented pastes. The relative recoveries of graphene-coated magnetic-sheet SPE were 83–102%, and the reusability was 20 times without significant loss of extraction recovery (< 4%).

Meanwhile, polydopamine-coated magnetic nanoparticles (PD-MNPs) were used as the adsorbent by preparation from amine-terminated magnetic nanoparticles (MNPs) and dopamine [98]. The extraction rates of AFB_1_ and AFG_2_ were 89.0% and 59.3%, and the limits of detection (LOD) were 0.0012 ng/mL for AFB_1_, AFB_2_, and AFG_1_, and 0.0031 ng/mL for AFG_2_.

Furthermore, the novel absorbent used an aptamer to functionalize magnetic agarosemicrospheres (MAMs) and bind to AFBs, then immobilized the AFB-aptamers on MAMs via a biotin–streptavidin interaction. The LOD values were 25 pg/mL for AFB_1_ and 10 pg/mL for AFB_2_ under reasonable conditions [99].

### 5.2. Aptamer Probes Techniques

Aptamers are specific affinity reagents with small sequences of oligonucleotides that can be screened for in vitro [100] and mimic antibodies by folding into complex 3D shapes that bind to specific targets [101]. Several works investigated aptamers in terms of designing them to construct various detection methods, such as the rapid detection of zearalenone in corn samples [102] and chloramphenicol determination [103]. Their best advantage is that they can be designed into numerous structures, including oligonucleotide probes to output signals directly.

An electrochemical aptasensor for aflatoxin B (AFB) detection using cyclic voltammetry (CV) and electochemical impedance spectroscopy (EIS) was developed with high sensitivity and selectivity. This aptasensor can be applied to detect AFB1 in white wine, peanuts, cashew nuts, and soy sauce, with a recovery of 85–100% [104].

In order to amplify the signal of aptamer probes, a novel AFB_1_ aptamer probe was studied using complementary DNA as a signal generator for amplification using real-time quantitative polymerase chain reaction (PCR) [105]. Many amplification strategies were adopted, involving complex enzyme catalysis and precise temperature control, which hindered the application of on-the-spot aflatoxin detection. Xia et al. proposed an aptamer probe design strategy to achieve AFB_1_ detection with enzyme-free amplification and homogeneous reactions [106], which was applied successfully in broad bean paste and peanut oil within one minute, which is the quickest detection of AFB_1_ so far.

Besides detecting AFBs, the aptamer was regarded as a high-affinity, specific method to detect OTA [107]. For example, the detection platform for OTA based on fluorescence aptamers was established, and exonuclease III (Exo III) was used to amplify signals and gold nanoparticles (AuNPs) to reduce fluorescence [108]. This aptasensor was confirmed to have excellent selectivity, with a limit of detection of 4.82 nM. Methods based on homogeneous electrochemical sensing protocol using a thionine–aptamer/graphene nanocomplex were also developed. The electronic signal starts off weak, but after making a connection with OTA, the thionine–aptamer/OTA generates an electrochemical signal using to detect OTA contamination in foods [109]. Moreover, biosensors using core–satellite nanostructures were applied to quantify mycotoxins, whereby the chiral intensity weakened by increasing OTA [110].

In recent years, a highly selective method integrating the electrochemistry and chemiluminescence of aptamers, electrochemiluminescence (ECL), has been widely used. Taking advantage of cadmium sulfide semiconductor quantum dots (CdS QDs) and DNA walkers, a novel technique based on the ECL aptasensor was confirmed to be a highly sensitive piece of equipment to quantify OTA during production, with a detection limit of 0.012 nM [111].

### 5.3. On-Site Test

Immunoassays are conventional techniques used AFB_1_ pretreatment in many foods. Due to this process being time-consuming and complex, many novel methods based on immunoassaying were developed. Immunomagnetism for pretreatment and enzyme-linked immunosorbent assay (ELISA) for quantification were established to detect AFB_1_ in soybean pastes [112]. These methods combined the active superparamagnetic beads and anti-AFB_1_ monoclonal antibodies to form immunomagnetic beads (IMBs). The magnetic beads coupled with the antibodies bind to a target material via an affinity reaction. The average recovery was between 0.5 and 7 µg/kg, or 83.6%~104%, and the relative standard deviation was 4.2%~11.7%.

A rapid on-site test for a wide range of aflatoxins in fermented soybean pastes is necessary. A novel strip test was developed based on lateral flow immunoassay, in which 3B6 was used to detect aflatoxins B_1_, M_1_, G_1_, G_2_, and B_2_, which belonged to monoclonal antibodies. The colloidal gold particles coated with 3B6 were considered to be the detector reagent, whereas the AFB_1_–BSA conjugate was regarded as a competitor reagent. This method was applied as a rapid and cost-effective screening tool, which had a limit of detection (0.5 µg/kg) within 10 min [113].

### 5.4. Molecular-Based Techniques

In recent years, molecular-based techniques such as PCR have been widely applied to detect *B. cereus* through the amplification of specific cereulide genes [114]. However, post-PCR analysis and gel electrophoresis assays increase the possibility of cross-contamination [115]. Furthermore, quantitative real-time PCR (qPCR), which uses intercalating dyes such as SYTO9 or dual-labeled fluorescence probes, offers simultaneous amplification and target-gene detection, which traditional PCR does not [116]. Many works were based on qPCR. For instance, the multiplex qPCR technique was developed to detect different pathogens simultaneously, including *B. cereus* in a variety of foods. Based on targeting various specific genes as indicators in a single reaction, the multiplex qPCR assay showed high efficiency and accuracy with a detection limit of 10^3^ CFU/g for every pathogen [117]. 

Nevertheless, recent studies demonstrated the limitations of PCR-based methods, which cannot identify DNA from cells, regardless of viability. DNA amplification in nonviable cells leads to overestimation of bacterial counts [118]. Some researchers used propidium monoazide (PMA) as a nucleic acid to overcome this limitation by combining PMA with qRNA to detect and quantify *B. cereus* in ultra-high-temperature (UHT) milk samples, where the detection limit was 7.5 × 10^2^ CFU/mL. Due to cell apoptosis during the UHT process, the detection percentage of PMA–qPCR was 32.6% compared with 57.8% of traditional qPCR, indicating that the application of PMA treatment prevented the overestimation of *B. cereus* in food [119].

However, the techniques mentioned above either require dual-labeled probes or a separate probe, which are very expensive. A novel pentaplex RT-PCR high-resolution melt-curve assay using DNA intercalating dye SYTO9 was developed [120], which has the ability to quantify four enterotoxin genes, namely nheA, entFM, cytK, and hblD, simultaneously, as well as ces, thereby allowing a wide range of detection of *B. cereus*. Furthermore, the detection limit in foods is 10^1^ CFU/g with seven hours of enrichment. This investigation revealed an economical alternative method for melt-curve analysis and probe-based real-time detection technology.

Although PCR assays are sensitive enough and widely applied in cereulide detection, the fact that *B. cereus* strains have widely distributed genes presents a limitation [60]. A recent study demonstrated that expanding the number of genes though whole genome sequencing (WGS) was considered to be a more powerful technique than PCR-based methods [121]. Therefore, using WGS as pre-screening measure and BTyper as a data analysis tool would be a useful method to detect toxin genes in mixed cultures.

## 6. Conclusions

Three kinds of mycotoxins, namely, aflatoxins, ochratoxins, and cereulide, are often detected in fermented pastes. Different harvest stages, appropriate drying processes, relatively low temperatures, and organic acid pretreatments are the primary influential factors which will significantly help to guide fermentation industries to effectively avoid mycotoxin contamination. This review provided feasible suggestions for effective control, such as biocontrol agents and rapid detection methods throughout the whole production process of fermented pastes.

## Figures and Tables

**Figure 1 toxins-12-00078-f001:**
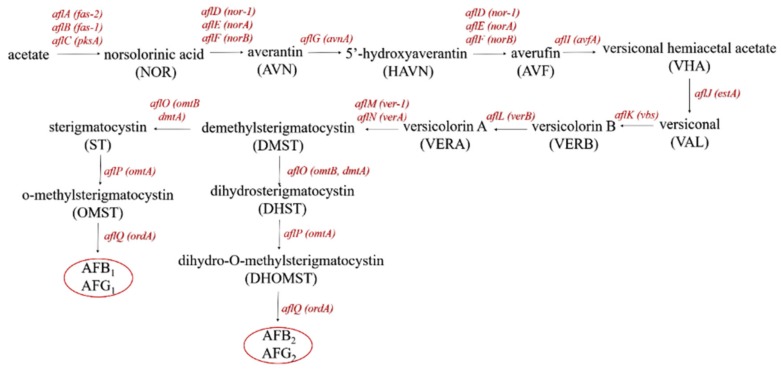
The conversion pathway of aflatoxin production: aflA: fatty acid synthase α; aflB: fatty acid synthase β; alfC: polyketide synthase; aflD: reductase; aflE: NOR-reductase; aflF: dehydrogenase; aflG: P450 monooxygenase; aflH: alcohol dehydrogenase; aflI: oxidase; aflJ: esterase; aflK: VERB synthase; aflL: desaturase; aflM: dehydrogenase; aflN: monooxygenase; aflO: O-methyltransferase B; aflP: O-methyltransferase A; aflQ: oxidoreductase.

**Figure 2 toxins-12-00078-f002:**
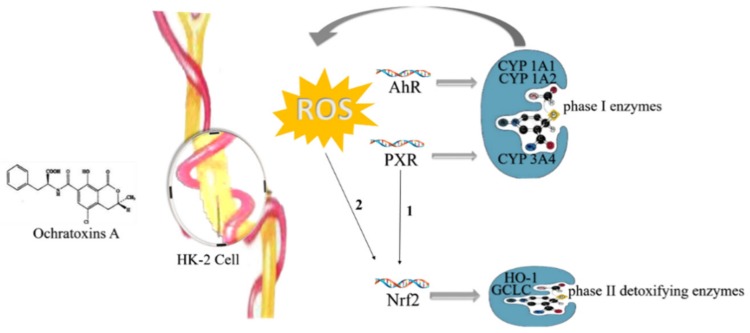
Pathogenic mechanism of ochratoxin A in Human Kidney (HK)-2 cells: ROS: reactive oxygen species; AhR: aryl hydrocarbon receptor; PXR: pregnane X receptor; Nrf2: NF-E2-related factor 2; HO-1: heme oxygenase-1.

**Figure 3 toxins-12-00078-f003:**
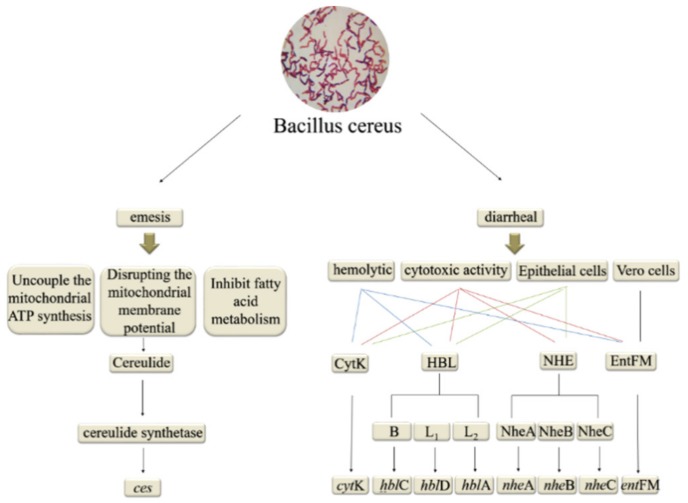
The enzymes and genes involved in *Bacillus cereus* toxin pathogenesis.

**Figure 4 toxins-12-00078-f004:**
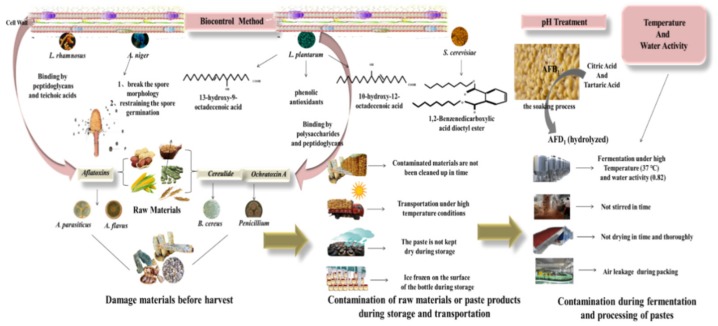
Mycotoxin production mechanism and control measures in fermented pastes.

**Table 1 toxins-12-00078-t001:** The list of the 18 identified aflatoxin-producing species.

Species	Aflatoxins	Reference
*Aspergillus aflatoxiformans*	B_1_, B_2_, G_1_, G_2_	[14]
*Aspergillus arachidicola*	B_1_, B_2_, G_1_, G_2_	[16]
*Aspergillus austwickii*	B_1_, B_2_, G_1_, G_2_	[14]
*Aspergillus cerealis*	B_1_, B_2_, G_1_, G_2_	[14]
*A. flavus*	B_1_, B_2_	[17]
*Aspergillus luteovirescens*	B_1_, B_2_, G_1_, G_2_	[16]
*Aspergillus minisclerotigenes*	B_1_, B_2_, G_1_, G_2_	[16]
*Aspergillus mottae*	B_1_, B_2_, G_1_, G_2_	[18]
*A. nomius*	B_1_, B_2_, G_1_, G_2_	[19]
*Aspergillus novoparasiticus*	B_1_, B_2_, G_1_, G_2_	[20]
*A. parasiticus*	B_1_, B_2_, G_1_, G_2_	[21]
*Aspergillus pipericola*	B_1_, B_2_, G_1_, G_2_	[14]
*Aspergillus pseudocaelatus*	B_1_, B_2_, G_1_, G_2_	[22]
*Aspergillus pseudonomius*	B_1_	[22]
*Aspergillus pseudotamarii*	B_1_, B_2_	[22]
*Aspergillus sergii*	B_1_, B_2_, G_1_, G_2_	[18]
*Aspergillus togoensis*	B_1_	[23]
*Aspergillus transmontanensis*	B_1_, B_2_, G_1_, G_2_	[18]

**Table 2 toxins-12-00078-t002:** The pathogenicity mechanism of mycotoxins.

Mycotoxins	Diseases	Aim Organ	Related Enzymes
AFB_1_	Human hepatocellular carcinoma	Human liver	CYP_1_A_2_, CYP_3_A_4_
Ochratoxin A	Balkan Endemic Nephropathy, Tunisian Nephropathy	HK-2 cells	CYP_1_A_1_, CYP_1_A_2_, CYP_3_A_4_
Ochratoxin A	Parkinsonism, Alzheimer’s disease	Human brain	OGG_1_
Cereulide	Gastrointestinal diseases	Intestinal epithelial cells	CytK, Hb_l_, Nhe, EntFM
Cereulide	Emetic illness	Mitochondria	Cereulide synthetase

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
