# Peer review of "Predominant Mycotoxins, Pathogenesis, Control Measures, and Detection Methods in Fermented Pastes"

_toxins, 2020, doi:10.3390/toxins12020078_

Round 1

Reviewer 1 Report

This is an interesting review of mycotoxins in different fermented pastes. I have no major remarks, just a few corrections, and suggestions:

Key contributions should be re-written to explain what is the actual contribution of this paper to the scientific and professional community. 

Line 20: correct "fermentd"

Line 21: please rephrase "make people sick", this sounds very common.

Line 210: please rephrase the subtitle

My major remark is that the English language should be proofed because some of the sentences could be improved. 

Author Response

This is an interesting review of mycotoxins in different fermented pastes. I have no major remarks, just a few corrections, and suggestions:

1). Key contributions should be re-written to explain what is the actual contribution of this paper to the scientific and professional community. 

Response: Thank you for your suggestion. The purpose of this review is to summarize the source, the pathogenicity mechanism, the control method and the novel detection technology of the mycotoxins. The actual contribution of this review was added in the Key Contribution section.

2). Line 20: correct "fermentd"

Response: Sorry for this mistake, and the word “fermentd” was revised to “fermented”.

3). Line 21: please rephrase "make people sick", this sounds very common.

Response: Thank you for your suggestion. The rephrase was revised to be more concise and academic. The sentence was revised to “The mycotoxins are always found in fermented paste which can lead to serious illness”.

4). Line 210: please rephrase the subtitle

Response: The subtitle was revised to “Physicochemical control methods”.

5). My major remark is that the English language should be proofed because some of the sentences could be improved.

Response: Thank you for your suggestion. We revised the whole manuscript carefully to avoid language errors. In addition, we asked several colleagues who are skilled authors to check the English. We believe that the language is now acceptable for the next review process.

Reviewer 2 Report

This paper should be presented as "Perspective" and not as "Review". Some introductive lines on food quality, food safety  and nutraceutical value of food along food chain should be added and related references inserted such as Durazzo Lucarini, 2018. A current shot and re-thinking of antioxidant research strategy. Brazilian Journal of Analytical Chemistry5(20), pp. 9-11.

The aim of this research should be explained.

Lines 51-52 should be enlarged.

Figure 1 and Figure 2 should be better discussed in the text.

The subparagraph "Cereulide" should be implemented. 

A Table describing the main Pathogenicity mechanism of mycotoxins should be added.

A Table describing the methods to control and manage mycotoxins should be added.

Proper conclusion describing the innovative character of this research should be added.

Author Response

1). This paper should be presented as "Perspective" and not as "Review". Some introductive lines on food quality, food safety and nutraceutical value of food along food chain should be added and related references inserted such as Durazzo Lucarini, 2018. A current shot and re-thinking of antioxidant research strategy. Brazilian Journal of Analytical Chemistry5(20), pp. 9-11.

Response: Thank you for your suggestion. Some introductive lines on food quality, food safety and food nutraceutical value was added, and the reference “current shot and re-thinking of antioxidant research strategy” was added in the introduction. Thank you.

2). The aim of this research should be explained.

Response: Thank you for your suggestion. The purpose of this review is to summarize the source, the pathogenicity mechanism, the control method and the novel detection technology of the mycotoxins. The aim of this review was added in the Key Contribution section.

3). Lines 51-52 should be enlarged.

Response: Thank you and the sentence was enlarged to “Aflatoxins are ubiquitously found in cereals, milk, tree nuts and oilseeds and considered as a group of extremely toxic metabolites which are produced…” which can be understand by readers more clearly.

4). Figure 1 and Figure 2 should be better discussed in the text.

Response: Thank you very much for this suggestion. The figures are hard to understand before, and some more discussions and descriptions were added in the right positions which can be better understand now. Figure 2 was removed to the “3. methods to control and manage mycotoxins” section.

5). The subparagraph "Cereulide" should be implemented. 

Response: Thank you. The subparagraph of “Cereulide” was implemented, and some description sentences were added in the section.

6). A table describing the main Pathogenicity mechanism of mycotoxins should be added.

Response: Thank you very much. Table 2 describing the main pathogenicity mechanism of mycotoxins was added.

7). A table describing the methods to control and manage mycotoxins should be added.

Response: Thank you for your constructive suggestion. The methods to control and manage mytotoxins was was clearly described and showed in Figure 4.

8). Proper conclusion describing the innovative character of this research should be added.

Response: Thank you for your suggestion. Different harvest stages, appropriate drying process, relatively low temperature and organic acid pretreatment are primary influence factors, which will significantly guide the fermented industries to avoid mycotoxins contamination effectively. This review provides feasible suggestions for effective controlling such as the biocontrol agent, and rapid detection methods during the whole production process of fermented pastes.

Round 2

Reviewer 2 Report

The authors have improved the manuscript that it is now suitable for publication